# Co-crystal structures of the fluorogenic aptamer Beetroot show that close homology may not predict similar RNA architecture

Luiz F. M. Passalacqua [1], Mary R. Starich [1], Katie A. Link [1], Jiahui Wu [2,3], Jay R. Knutson[1], Nico Tjandra [1], Samie R. Jaffrey [2] & Adrian R. Ferré-D'Amaré [1] ✉

Beetroot is a homodimeric in vitro selected RNA that binds and activates DFAME, a conditional fluorophore derived from GFP. It is 70% sequence-identical to the previously characterized homodimeric aptamer Corn, which binds one molecule of its cognate fluorophore DFHO at its interprotomer interface. We have now determined the Beetroot-DFAME co-crystal structure at 1.95 Å resolution, discovering that this RNA homodimer binds two molecules of the fluorophore, at sites separated by ~30 Å. In addition to this overall architectural difference, the local structures of the non-canonical, complex quadruplex cores of Beetroot and Corn are distinctly different, underscoring how subtle RNA sequence differences can give rise to unexpected structural divergence. Through structure-guided engineering, we generated a variant that has a 12-fold fluorescence activation selectivity switch toward DFHO. Beetroot and this variant form heterodimers and constitute the starting point for engineered tags whose through-space inter-fluorophore interaction could be used to monitor RNA dimerization.

Fluorescence turn-on aptamers are in vitro-evolved RNAs that strongly activate their cognate conditional fluorophores. These RNAs have emerged as counterparts to fluorescent proteins, and have been widely used in applications ranging from live imaging of cellular RNAs to small-molecule sensors[1–9]. Structure determination of several fluorogenic RNAs revealed diverse architectures with idiosyncratic solutions to recognize and then restrain the photoexcited fluorophores in a planar conformation[1,3,10]. In particular, G-quadruplexes, which provide a thermodynamically stable, planar binding surface, appear to be overrepresented in fluorogenic RNAs compared to small-molecule-binding aptamers generally[1,3,11]. Several fluorogenic aptamers, including Broccoli, Chili, Spinach, and Squash (refs. 12–15), have been evolved to activate small molecule fluorophores derived from the intrinsic fluorophore of green fluorescent protein (GFP), such as DFHBI (**1**) and DFHO (**2**) (Fig. 1a). All of these aptamers function as monomers, except for Corn[16], which homodimerizes, and binds one molecule of DFHO at its interprotomer interface[17,18]. Each of the two Corn protomers folds into a stem capped by a G-quadruplex, and the planar surfaces of the G-quadruplexes of the two protomers associate to form the interprotomer interface and fluorophore-binding site[17,18]. It was suggested that if a heterodimeric Corn variant could be engineered, it could be used to report on co-localization or dimerization of cellular RNAs, in effect, as an analogue of split GFP (ref. 17). However, because the dimer interface and the fluorophore binding site of Corn are the same, engineering efforts have been fruitless, with mutations that affect dimerization also negatively impacting fluorescence activation[18].

Beetroot is a recently discovered[19] fluorogenic aptamer RNA that binds and activates fluorescence of the extended-conjugation GFP-derived fluorophore, DFAME (**3**) (Fig. 1a). The Beetroot-DFAME complex exhibits a large Stokes shift, and its red-shifted emission provides

[1]Biochemistry and Biophysics Center, National Heart, Lung, and Blood Institute, National Institutes of Health, Bethesda, MD, USA. [2]Department of Pharmacology, Weill-Cornell Medical College, Cornell University, New York, NY, USA. [3]Present address: Department of Chemistry, Binghamton University, Binghamton, NY 13902, USA. ✉e-mail: adrian.ferre@nih.gov

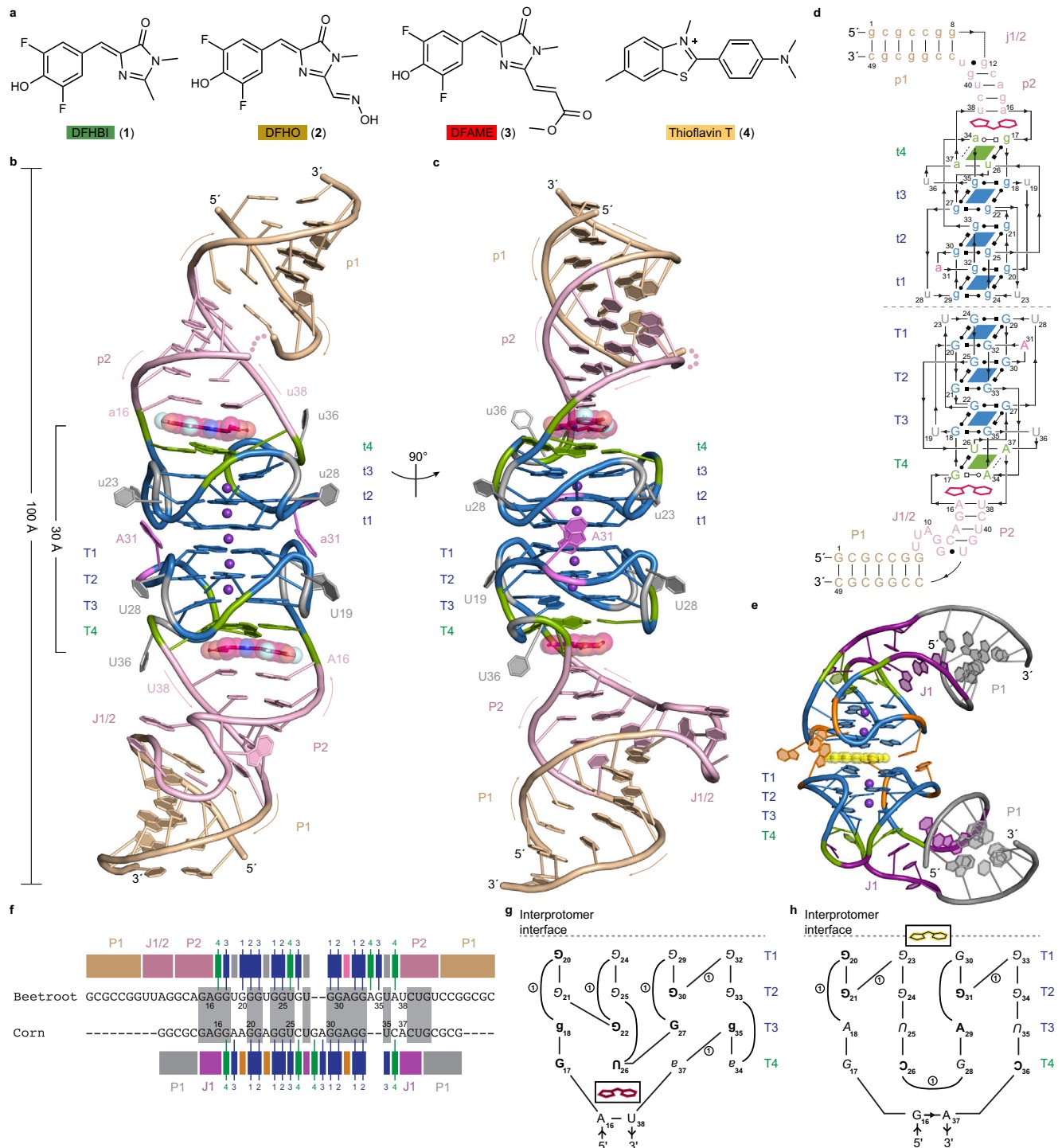

**Fig. 1 | Overall structure of the Beetroot fluorescence turn-on aptamer in complex with DFAME. a** Chemical structures of the fluorophores DFHBI (**1**), DFHO (**2**), DFAME (**3**), and ThT (**4**). **b** Cartoon representation of the Beetroot-DFAME homodimeric complex. Arrows indicate 5′ to 3′ chain direction, and purple spheres represent K⁺. The bound DFAME molecules are shown in ball-and-stick representation with translucent spheres. **c**, Orthogonal view of **b**. **d** Secondary structure representation of Beetroot color-coded as in **c**. Lines with embedded arrowheads and Leontis-Westhof symbols[41] denote connectivity and base pairs, respectively. **e** Cartoon representation of the homodimeric fluorogenic aptamer Corn-DFHO complex[17] (PDB: 5BJO). **f** Sequence alignment of the aptamers Beetroot and Corn, both numbered according to the Beetroot register. Nucleotides conserved in sequence are boxed in gray, while structural elements are denoted by rectangles colored as in **c** and **e**. Nucleotides that form the four quartets of each RNA are numbered in color. **g** Graphical G-quartet schematics (ref. [11]) for Beetroot and **h** Corn, in Beetroot numbering. Each row represents the nucleotides of a quartet tier, and columns indicate nucleotide stacks. Upper-case, lower-case, bold, and Italic letters denote *anti*, *syn*, 2′-endo and 3′-endo nucleotides, respectively; upside-down letters denote strand polarity inversion with respect to the 5′-most nucleotide in the scheme. Lines connecting nucleotides are loops and bulges, with the number of nucleotides indicated within a circle.

advantages compared to earlier fluorogenic aptamers for cellular imaging[19]. Biochemical characterization indicates that Beetroot forms a stable homodimer independent of ligand binding, with an RNA:RNA dissociation constant ($K_d$) of less than 1 nM (ref. [19]). The functional cores of Beetroot and Corn have a high degree of sequence identity (19 identical nucleotides out of 27, with one gap in each sequence). Despite this, the two aptamers do not form mixed RNA heterodimers[19], and have different fluorophore selectivity (Beetroot and Corn bind preferentially to DFAME and DFHO, respectively). The dimerization orthogonality and divergent fluorophore selectivity of Beetroot and Corn is unexpected, given their high sequence identity, but suggests that these two aptamers could potentially be the starting point for the future development of fluorogenic aptamer systems that exploit the combinatorial possibilities of oligomeric RNAs. Such engineering of quaternary structure[20] remains a challenging frontier in functional RNA design.

To elucidate how two closely related RNA sequences can nonetheless have orthogonal dimerization selectivity, and as a starting point for engineering the quaternary structure of a fluorogenic RNA, we have now determined the co-crystal structure of the Beetroot–DFAME complex at 1.95 Å resolution. Remarkably, despite sharing structural motifs with Corn (consistent with sequence similarity), Beetroot dimer arranges these motifs in a distinctly different architecture: instead of binding one fluorophore at its interprotomer interface, it binds two DFAME molecules ~30 Å apart. We have also determined the structure of the complexes of Beetroot with DFHO and with Thioflavin-T (**4**) (ThT, Fig. 1a), to shed light on the structural basis for its fluorophore selectivity. ThT is a promiscuous G-quadruplex binder[21,22] bound and activated by Corn at its interprotomer interface[18]. We find that Beetroot dimer binds two molecules of DFHO or ThT, in the same distant binding sites it uses for DFAME. Through structure-guided mutagenesis, we developed a Beetroot variant that preferentially activates DFHO, and used this to engineer a heterodimeric Beetroot, which activates one molecule each of DFAME and DFHO, in the corresponding cognate sites. Our structure-guided engineering of the architectural features of Beetroot underscores the potential of exploiting RNA quaternary structure for generating new molecular tools.

## Results

### Beetroot is a quasi-symmetric homodimer

An RNA construct comprising the conserved Beetroot core, extended with a 7 bp terminal helix (Supplementary Table 2), in complex with DFAME (**3**), produced brightly fluorescent crystals (Supplementary Fig. 1). Its structure was determined by molecular replacement using two copies of the two all-guanine tiers of the G-quadruplex of the Corn-DFHO co-crystal structure[17] as a search model and refined at 1.95 Å resolution (Supplementary Fig. 2, Supplementary Table 1, Methods). All our other Beetroot complex co-crystals are isomorphous with those of the DFAME complex, and all contain a dimer of the corresponding RNA-fluorophore complex in the crystallographic asymmetric unit.

The Beetroot–DFAME complex (Fig. 1 b–e) is a quasi-symmetric[20] homodimer, in which some equivalent nucleotides of the two RNA protomers adopt non-identical conformations (hereafter, the protomers are denoted A and B, and residues and structural elements of the two are identified with upper- and lower-case letters, respectively). Each protomer comprises two duplex segments (P1 and P2) separated by an irregular junction (J1/2) and a mixed connectivity quadruplex. The latter has three canonical G-quartet tiers (T1, T2, and T3) and one mixed-composition (G•U•A•A) tier (T4). Excluding J1/2, which is disordered in protomer B, the two protomers superimpose with a root-mean square difference (r.m.s.d.) of 1.35 Å (for 45 C1′ atom pairs). The structural elements of the dimer form a 100 Å-tall continuous coaxial stack, with T1 and t1 of the two RNAs stacking on each other to form the interprotomer interface.

### The Beetroot dimer binds two fluorophore molecules

Our co-crystal structure revealed that the Beetroot homodimer binds two DFAME molecules, which are separated by ~30 Å. Each fluorophore stacks between the face of the G-quartet distal from the dimer interface, and the adjacent duplex P2. This is unexpected because Corn, with which Beetroot shares 70% sequence identity, instead folds into a homodimer that binds to one molecule of its cognate fluorophore that forms an integral part of its interprotomer interface. No electron density that could correspond to additional bound DFAME molecules was observed in the Beetroot dimer interface or elsewhere in the crystallographic unit cell (not shown). Comparison of the structures and sequences of Beetroot and Corn shows how sequence elements that are conserved between the two RNAs (Fig. 1f) are employed to construct different structural motifs in the two related aptamers. This is evident when the aligned sequences of the two RNAs are annotated with their corresponding structural roles (Fig. 1f; see Supplementary Fig. 3 for alternative residue numbering). Thus, the first residue that forms part of the quadruplex (numbered G17 in both sequences in Fig. 1f) are offset by one position in the sequence alignment, and the first loop residue in Corn (A19) aligns in sequence with Beetroot residue G20, which forms part of the bottom (T4 or t4) G-quartet of the latter RNA. This discrepancy between the structural roles of nucleotides that align between the Beetroot and Corn sequences persists through the quadruplex element, which is overall one nucleotide shorter in Corn (Fig. 1f, Supplementary Fig. 3a). Importantly, the G-quadruplexes of the two RNAs are not simply offset by one nucleotide; the patterns of bulges and loops, and of stacking of nucleotides, are distinctly different in the two RNAs (Fig. 1g, h; Supplementary Fig. 3b, c).

The structures of the two fluorophore-binding sites of Beetroot are similar (r.m.s.d.=0.89 Å for all non-hydrogen atoms in Fig. 2a). DFAME binds the RNA with its two rings and its methylacrylate substituent coplanar, sandwiched between the G17•U26•A34•A37 (T4) tetrad and the terminal A16•U38 Watson-Crick pair of P2 (Fig. 2a–c). Compared to a canonical G-quartet, T4 is unusual in three ways. First, unlike the canonical G-quartet tiers (T1, T2, and T3), that coordinate $K^+$ ions at their 4-fold axis, T4 lacks a crystallographically ordered metal ion (or even a water molecule) in the corresponding position. Second, the cyclic hydrogen bonding that characterizes canonical G-quartets is interrupted in T4. While the bases of G17 and U26, U26 and A37, and A34 and G17 all share hydrogen bonds between each other (forming Watson-Crick•Hoogsteen, Watson-Crick, Watson-Crick•Sugar interactions, respectively), no hydrogen bonds are present between the bases of A34 and A37. Third, T4 is highly buckled, such that no two bases are coplanar. The phenyl ring of DFAME stacks between G17 and A16, while its methylacrylate portion between U26 and A37 of T4 of U38 of P2 (Fig. 2a–c). Remarkably, the fluorophore makes no hydrogen bonds to the RNA. The binding site is comparatively open, and the RNA only buries 40% of the total 523 Å$^2$ solvent-accessible surface area of DFAME. Consistent with this, several crystallographically ordered water molecules are present around the exposed sides of the bound fluorophore; however, at the present resolution limit, no water molecules appear to bridge DFAME and the RNA.

### The Beetroot quadruplex dimer core

Co-axial stacking of the two Beetroot protomers results in an eight-tiered quadruplex spanning the dimer interface. Five octacoordinated $K^+$ ions lie on its four-fold symmetry axis, equidistant from the planes of successive canonical G-quartets (Figs. 1b, c and 2d). The central $K^+$ ion spans the interprotomer interface. The G-quadruplex of each protomer has five extrahelical nucleotides (four uridines and one adenosine, Fig. 1g), with the adenosine (A31 or a31) being the sole nucleobase (other than the T1 and t1 guanines) that interacts with the other protomer. In addition, at the current resolution limit, numerous well-ordered water molecules appear to bridge the two RNA

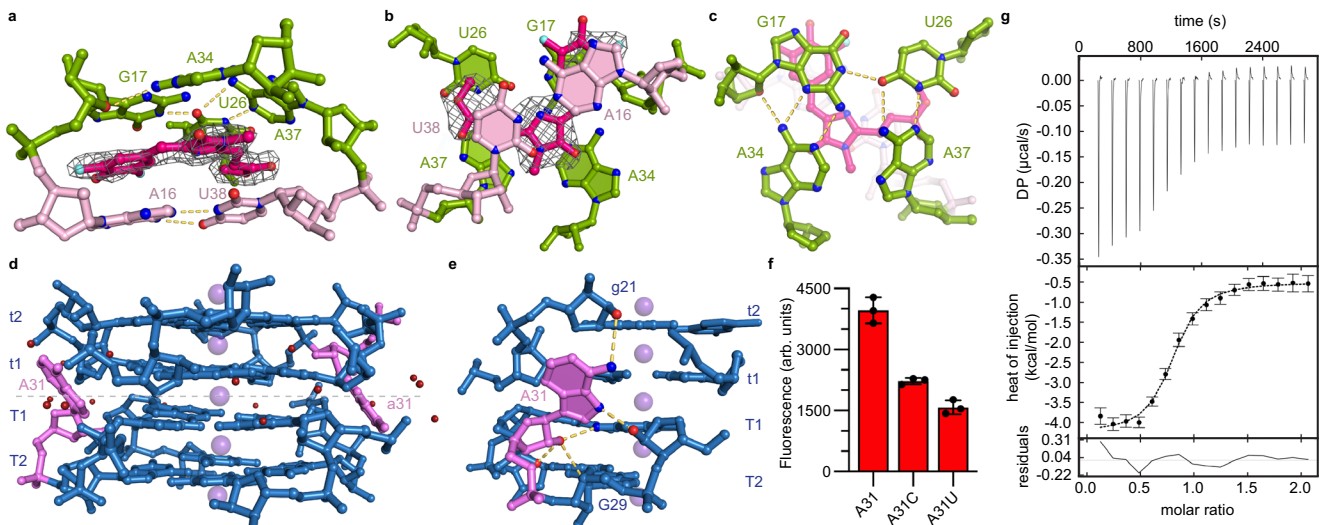

**Fig. 2 | Features of the Beetroot-DFAME binding site and the interprotomer interface. a** View of the binding site with bound DFAME sandwiched between the T4 tetrad and the terminal A16•U38 Watson-Crick pair of P2, colored as in Fig. 1. Gray mesh depicts the $|F_o| - |F_c|$ electron density map before building DFAME fluorophore, contoured at 3.0 standard deviations above mean peak height (σ). Yellow dashed lines denote hydrogen bonds. **b** Orthogonal view of **a** the binding site of Beetroot–DFAME from the direction of P2. **c** Orthogonal view of **a** the binding site from the direction of the interprotomer interface and with the electron density map omitted. **d** View of the interprotomer interface of Beetroot with the top two G-quartets tiers (T1 and T2, and t1 and t2) and the adenosine that makes interprotomer interaction (A31 and a31). The five octacoordinated K⁺ ions and water molecules (purple and red spheres, respectively) are depicted. Gray dashed line denotes the dimer interface. **e** Detail of A31 and its interaction with both protomers. **f** Fluorescence activation of DFAME by wild-type and A31 mutants (mean±s.d., $n = 3$ – $n$ denotes independent sample replicate). **g** Baseline-corrected thermogram, non-linear least-squares fit, and residuals for Beetroot titration with DFAME (mean±s.d., $n = 1$ – $n$ denotes independent experiment). Refer to Supplementary Fig. 5a for triplicate data.

protomers (Fig. 2d). The interprotomer A31 interaction consists of a single hydrogen bond between its N6 and the 2'-OH of g21 (and, correspondingly, between the N6 of a31 and the 2'-OH of G21; Fig. 2e). The importance of this symmetrical interaction is borne out by site-directed mutagenesis, as the A31C and A31U point mutations decreased fluorescence by least 50% relative to wild-type (Fig. 2f).

The dissociation constant ($K_d$) of the Beetroot protomers, in the absence of fluorophore, has been reported to be less than 1 nM (ref. 19). When we examined the oligomeric state of fluorophore-free Beetroot in solution through size-exclusion chromatography (SEC), two distinct populations could be observed: an abundant and a sparsely populated species with small and large elution volumes ($Ve$), respectively (Supplementary Fig. 4). We followed the abundance the two populations over a 24 h period and found that the species with large $Ve$ decreased over time, (Supplementary Fig. 4), suggesting that the minor species corresponds to a misfolded monomer that dimerized over time. Analysis of Beetroot in solution using multi-angle light scattering coupled to SEC (SEC-MALS) indicated the molecular mass of the abundant, low $Ve$ species to be ~32 kDa, close to the calculated mass (32.5 kDa) of the homodimer (Supplementary Fig. 4).

### Fluorophore binding and activation by Beetroot
We examined the association of DFAME with Beetroot by isothermal titration calorimetry (ITC) (Fig. 2g and Supplementary Fig. 5). Fluorophore binding is exothermic, with an RNA protomer:fluorophore molar ratio of 1:1. Non-linear least-square fits of our thermograms to binding models with either two equivalent or two non-equivalent sites yielded fits of similar quality. Thus, our ITC results suggest that the two fluorophore binding pockets of the Beetroot dimer are equivalent, and that there is no thermodynamic linkage between them. We also examined Beetroot·DFAME association through fluorescence titrations. Analysis of the ITC and fluorescence experiments indicated DFAME dissociation constants of 0.47 μM and 1.55 μM (Fig. 2g and Supplementary Fig. 5), similar to what has been reported previously[19]. Additionally, the ITC experiments indicate that the association is both

entropically and enthalpically driven, with $\Delta H = -3.58$ kcal/mol and $T\Delta S = 5.08$ kcal/mol (at 21 °C).

We examined the importance of Beetroot structural elements for fluorophore activation by site-directed mutagenesis. First, mutations to T1 or T4 G-quartets, either to test if T1 is essential or whether making T4 more G-rich would yield a more stable molecule, abolished fluorescence (not shown, Supplementary Table 2). Second, we systematically mutagenized the A16•U38 base pair of P2. Examination of the ability of our mutants to induce DFAME fluorescence showed that A16C•U38G, A16U•U38G, and A16U•U38A to be the only active mutants, which nonetheless are considerably less fluorescent than the wild-type complex (Fig. 3a). Third, we examined the functional importance of J1/2 by deleting it. This mutant activates DFAME fluorescence comparably to wild-type, indicating that this extrahelical element does not play an important role in DFAME fluorescence turn-on (Supplementary Fig. 5).

### Structure of the DFHO and ThT complexes of Beetroot
We next examined the ability of Beetroot to activate three other fluorophores, DFHBI, DFHO and ThT, finding that the aptamer does not activate DFHBI, activates DFHO weakly, and strongly activates ThT (Fig. 3b). To provide a framework for understanding the ability of Beetroot to activate DFHO and ThT, we determined its structures in complex with these two fluorophores (Methods and Supplementary Table 1) at 2.55 Å and 2.1 Å resolution, respectively (Fig. 3c, d, Supplementary Table 1, Supplementary Figs. 2 and 6, Methods).

The overall structures of the Beetroot homodimer complexed with either DFHO or ThT are highly similar to that of the Beetroot–DFAME complex, with r.m.s. differences of 1.17 Å and 1.51 Å between the DFHO and DFAME complexes, and between the ThT and DFAME complexes, respectively (for 90 and 89 C1' atom pairs in each case and without the J1/2) (Fig. 3c, d and Supplementary Fig. 6). Both DFHO and ThT bind at opposite ends of the eight-tiered quadruplex at the dimer interface of Beetroot, in the same pockets occupied by DFAME and in near-planar conformations in both cases (Fig. 3c–h). Similar to DFAME, the phenyl

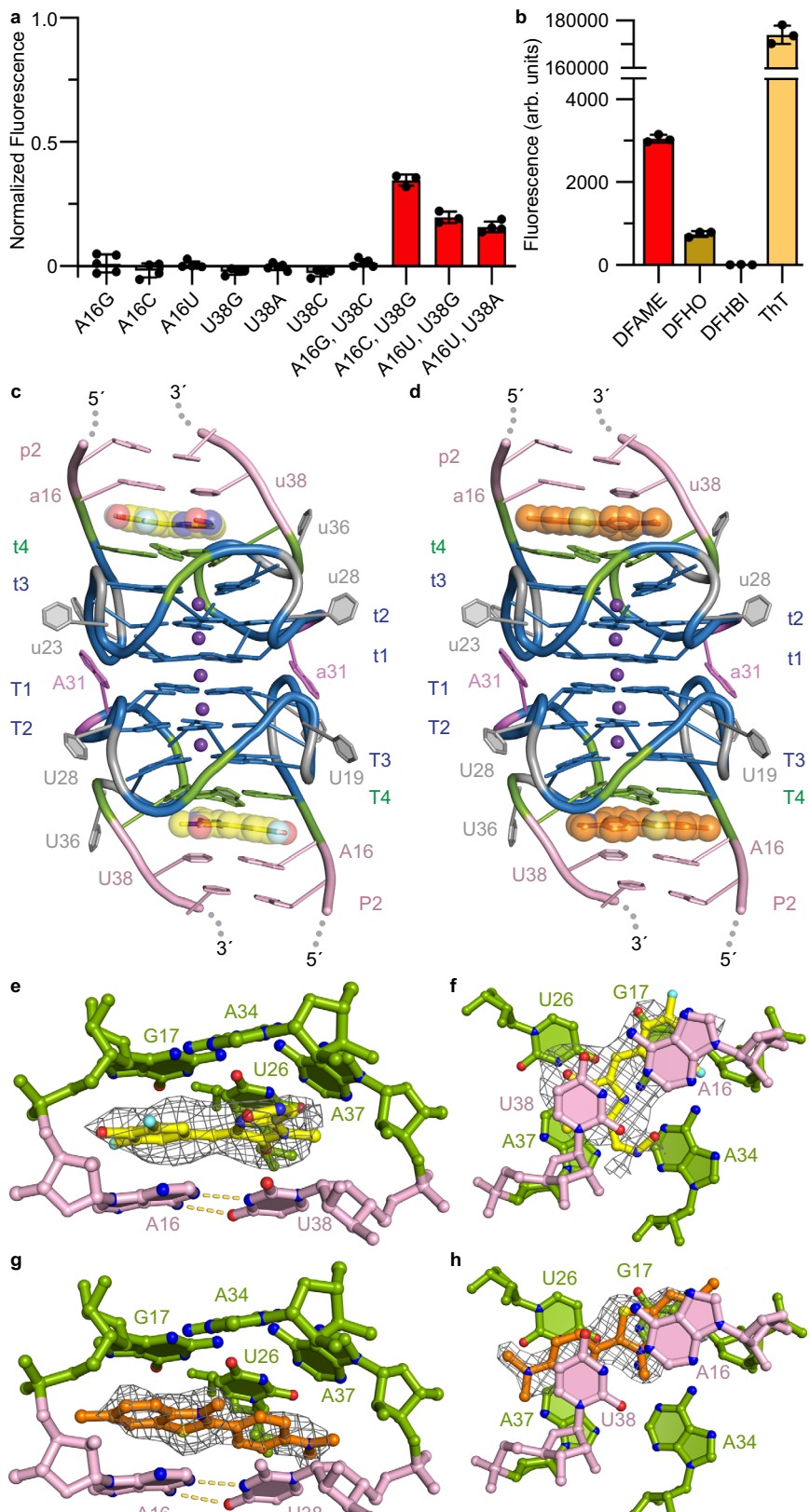

**Fig. 3 | Beetroot binding and activation of DFHO and ThT. a** DFAME fluorescence in the presence of Beetroot binding site mutants, normalized to wild-type (mean±s.d., $n = 5$ for A16G, A16U, and U38A; $n = 4$ for A16C, U38G, U38C, A16G,U38G, and A16U,U38A; $n = 3$ for A16C,U38G, and A16U,U38G – $n$ denotes independent sample replicate). **b** Fluorescence activation of four different fluorophores by Beetroot (mean±s.d., $n = 3$ – $n$ denotes independent sample replicate). **c** Cartoon representation of the core of the Beetroot–DFHO homodimer complex co-crystal structure. **d** Cartoon representation of the core of the Beetroot–ThT homodimer complex co-crystal structure. **e** Detail of the binding site with DFHO. **f** Orthogonal view of **e** the binding site of Beetroot–DFHO from the direction of P2. **g** Detail of the binding site with ThT. **h** Orthogonal view of **g** the binding site of Beetroot–ThT from the direction of P2. In **e**–**h** gray mesh depicts $|F_o| − |F_c|$ electron density map before building the fluorophores, contoured at 2.0 σ.

ring of DFHO is sandwiched between the nucleobases G17 of T4 and A16 of P2. Its carbaldehyde oxime substituent lies between the nucleobases of A34 and A37 of T4 and U38 of P2 (Fig. 3e, f). ThT binds with its benzothiazole between the nucleobases of G17 and A16, and its phenyl ring between U26 and A37 of the T4 and U38 of P2 (Fig. 3g, h). Comparable to Beetroot–DFAME, the RNA binding only buries 42% of the total 459 Å² solvent-accessible surface area of DFHO, and 38% of the total 485 Å² solvent-accessible surface area of ThT.

### Orthogonal fluorescence activation of DFHO by a Beetroot mutant

Because of the sensitivity of DFAME activation to the composition of the P2 base pair on which it stacks (A16•U38 in the wild-type; Fig. 3a), we hypothesized that the selectivity of Beetroot between DFAME and DFHO may be modulated by varying this base pair. We, therefore, examined all mutants of this base pair that would be expected to maintain Watson-Crick or wobble base pairing, for their ability to turn on DFHO fluorescence. Remarkably, the double mutation that converts this Watson-Crick pair into the wobble base pair U16•G38 (A16U, U38G) results in three times higher activation of DFHO, compared to the wild-type Beetroot (Fig. 4a). Moreover, this mutant (hereafter, Wobble Beetroot) activates DFAME fluorescence only 25% as much as

the wild-type (Fig. 3a). Thus, replacement of a single P2 base pair from A•U to U•G resulted in a ~ 12-fold switch of fluorescence activation from DFAME to DFHO.

To elucidate the structural basis of the selectivity switch from DFAME to DFHO of Wobble Beetroot, we next determined its co-crystal structure bound to DFHO at 2.85 Å resolution (Fig. 4b–d, Supplementary Table 1, Supplementary Fig. 2 and 6, Methods). The overall structure of Wobble Beetroot bound to DFHO is similar to those of the wild-type Beetroot fluorophore complexes (r.m.s.d. of 2.10 Å for C1' pairs between the Wobble Beetroot-DFHO and the wild-type Beetroot-DFHO complexes without J1/2; Fig. 4b, c and Supplementary Fig. 6). However, the DFHO bound to Wobble Beetroot has rotated ~ 30° on its plane when compared to its pose when bound to wild-type Beetroot (Fig. 4d, e). In addition, the conformation of the carbaldehyde oxime substituent is also reversed. Thus, this substituent stacks completely on the nucleobase of A34 in Wobble Beetroot, while it does not stack directly on any of the nucleobases of T4 in the wild-type (Fig. 4c–e). A34 does not participate in extensive stacking with either DFAME or ThT in the corresponding wild-type Beetroot complexes. Comparable to the other complexes of Beetroot–ligand here reported, the wobble Beetroot–DFHO binding only buries 37% of the solvent-accessible surface area of the ligand. Consistent with this, and as in our other

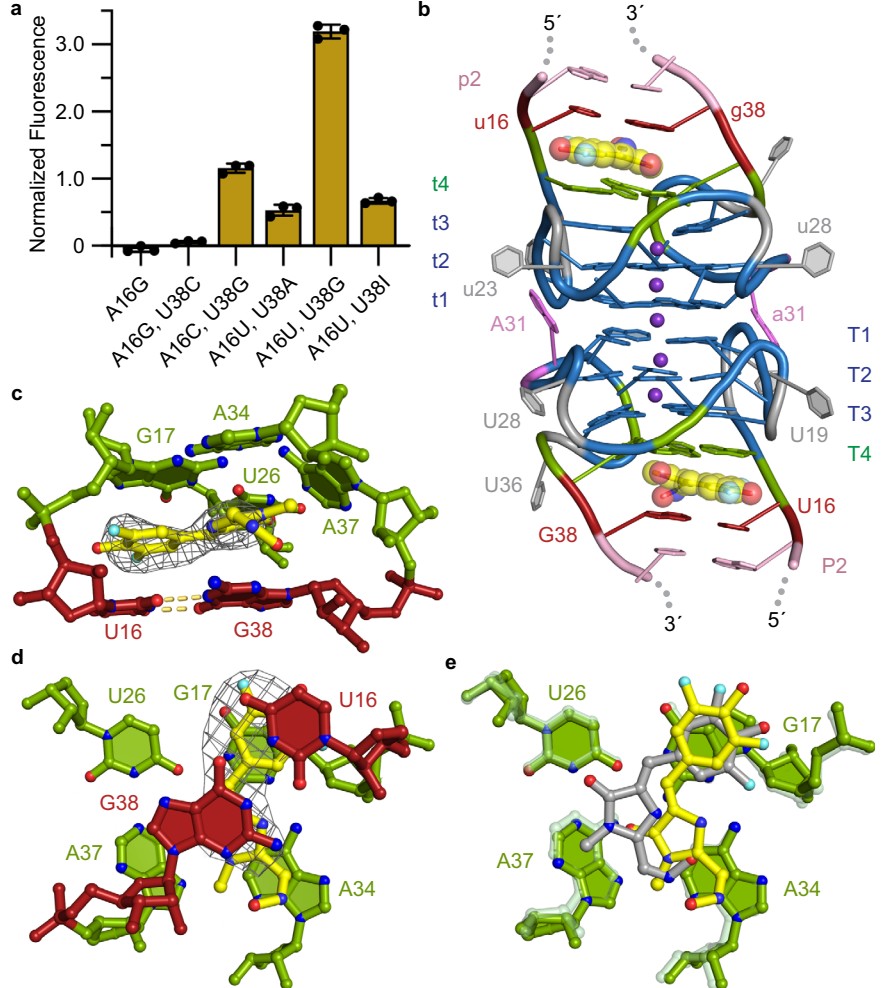

**Fig. 4 | Orthogonal fluorescence activation of DFHO by the Wobble Beetroot mutant. a** DFHO fluorescence in the presence of binding-site mutants of Beetroot, normalized to wild-type activated DFHO (mean±s.d., n = 3 – n denotes independent sample replicate). **b** Cartoon representation of the core of the Wobble Beetroot-DFHO co-crystal structure. The mutated U16•G38 wobble pair is colored red. **c** View of the Wobble Beetroot binding site with bound DFHO sandwiched between T4

tetrad and the U16•G38 wobble pair. Gray mesh depicts the |Fₒ| − |Fₒ| electron density map before building DFHO fluorophore, contoured at 2.0 σ. **d** Orthogonal view of **c** the binding site of Wobble Beetroot–DFHO from the direction of P2. **e** View of the superimposed binding sites of Wobble Beetroot–DFHO (nucleotides in green and ligand in yellow) and Beetroot–DFHO (nucleotides in light translucent green and ligand in gray) from the direction of P2 (base-pair of P2 omitted).

Beetroot complex structures, crystallographically ordered water molecules are present on both grooves of the P2 duplex, adjacent to the bound DFHO.

To examine qualitatively the effects on fluorophore dynamics of the mutations in Wobble Beetroot, we obtained $^{19}$F solution NMR spectra of DFAME and DFHO bound to wild-type and Wobble RNAs. The phenyl ring fluorines of the free DFAME resonate at −136.01 ppm. When bound to the RNA, the fluorine peak shifts downfield and broadens (Fig. 5a, b). The shift is more pronounced when DFAME binds to wild-type Beetroot (Δ = +0.05 and +0.02 ppm for wild-type and Wobble,

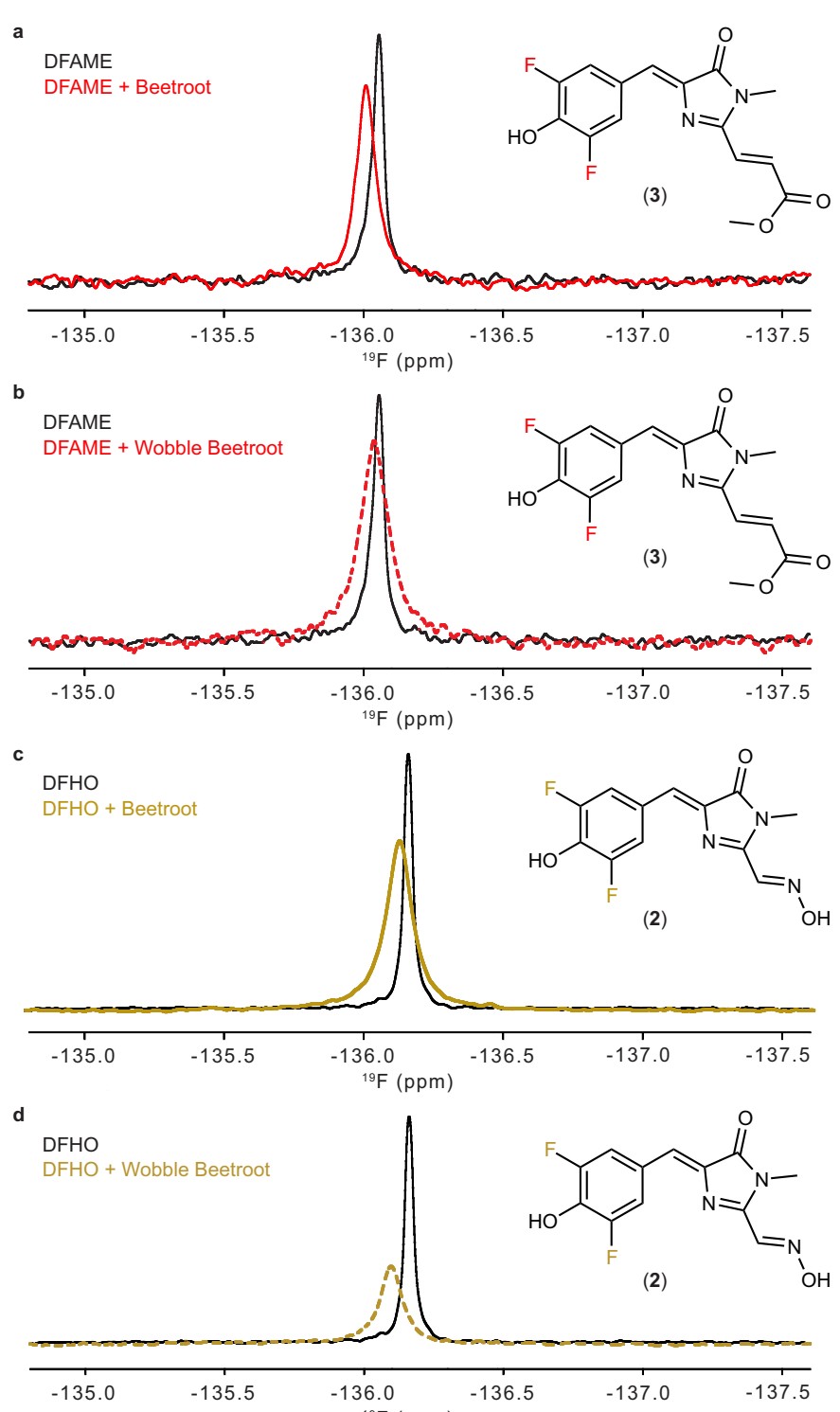

**Fig. 5 | Fluorophore dynamics of wild-type and Wobble Beetroot. a** $^{19}$F NMR spectra of DFAME free and bound to wild-type Beetroot and **b** Wobble Beetroot, in the region corresponding to the aryl fluorines. **c** $^{19}$F NMR spectra of DFHO free and bound to wild-type Beetroot and **d** Wobble Beetroot, in the same spectral region as in **a** and **b**. Insets indicate the fluorines of the fluorophores assigned to the observed resonances.

respectively). The fluorines of DFHO resonate at −136.16 ppm when the fluorophore is free in solution. When bound to the RNAs, its fluorine peak also shifts downfield and broadens (Fig. 5c, d). In this case, the broadening and the shift are more pronounced when DFHO binds to Wobble Beetroot (Δ = +0.03 and +0.07 ppm for wild-type and Wobble, respectively). The $^{19}$F nucleus is quite sensitive to its surroundings and when bound tightly, it can show substantial chemical shift perturbation due to environmental changes in hydrophobicity, charge and ring current effects. The modest change in chemical shift observed upon ligand association with Beetroot suggests loose binding, which may be due to the openness of the binding site that buries only ~ 40% of the fluorophores. Considering the small chemical shift changes observed (non-cognate pair Wobble Beetroot in the presence of DFAME), we obtained $^{19}$F solution NMR spectra of DFAME in the presence of a structured RNA that was not selected to bind DFAME. We chose the cleaved hammerhead ribozyme (HHR) that was used to generate the 5′-OH termini Beetroot. The HHR used for the NMR studies does not activate DFAME fluorescence (Supplementary Fig. 7). The $^{19}$F solution NMR spectra of DFAME in the presence of HHR exhibited a downfield shift (Δ = +0.02) comparable to that of DFAME in the presence of Wobble Beetroot (Supplementary Fig. 7). This suggests that non-specific interactions can cause chemical shift changes of this magnitude, and that the larger shifts observed with DFHO binding to Wobble Beetroot (Δ = +0.07 ppm) and DFAME biding to wild-type Beetroot (Δ = +0.05 ppm) are indicative of specific binding.

Dissociation constants derived from fluorescence experiments of wild-type−DFHO, Wobble Beetroot−DFAME, and −DFHO are ~ 9 μM, 5 μM and 7.5 μM, respectively (Supplementary Fig. 8), indicating that DFHO has only modestly higher affinity for Wobble Beetroot. Additionally, the $K_d$ of Wobble Beetroot for DFAME is smaller than its $K_d$ for DFHO, indicating that fluorescence activation does not correlate directly with binding affinity. We also determined fluorescence lifetimes for the DFAME and DFHO bound to wild-type and Wobble Beetroot (Supplementary Fig. 9). The wild-type Beetroot−DFAME and Wobble Beetroot−DFHO complexes exhibit similar lifetimes (2.2 ± 0.2 ns and 2.3 ± 0.4 ns, respectively), while the non-cognate complexes had shorter lifetimes (1.0 ± 0.3 ns and 1.8 ± 0.1 ns, for Beetroot−DFHO and Wobble Beetroot−DFAME, respectively). Overall, our $^{19}$F NMR spectra and fluorescence lifetime analyses are consistent with our crystallographic observations indicating that Wobble Beetroot selectively activates fluorescence of DFHO over DFAME.

### The Beetroot fold supports heterodimer formation

To examine if the Beetroot RNA fold can be repurposed to generate a heterodimer, we fused wild-type Beetroot to the F30 scaffold[23], to facilitate analysis by making one of the RNAs larger. When the F30-wild-type fusion was folded together with Wobble Beetroot, heterodimer formation could be detected by native polyacrylamide gel electrophoresis (Fig. 6a). The stained gels revealed the presence of three distinct bands, corresponding the F30-wild-type and Wobble homodimers and the heterodimer (Fig. 6b). Excision of the band with intermediate mobility and analysis by denaturing polyacrylamide gel electrophoresis of the eluted RNA confirmed it to be the heterodimer (Fig. 6c). To verify the stability of the heterodimer, the heterodimer was analyzed by native polyacrylamide gel electrophoresis 48 h after purification. Results showed no formation of homodimers, indicative of dissociation (Supplementary Fig. 10a).

To examine if the wild-type and Wobble protomers of the heterodimer retained their respective fluorophore selectivity, we measured their fluorescence spectra in the presence of either or both DFHO and DFAME. Both fluorophores can be excited at 470 nm. When only DFAME or DFHO was present, the emission spectra matched those corresponding to the homodimeric F30-wild-type Beetroot−DFAME and variant Wobble Beetroot−DFHO complexes, with peak emissions at ~ 620 nm and 550 nm, respectively (Fig. 6d and

Supplementary Fig. 10b). When both DFAME and DFHO were present in solution, emission for both activated DFHO and DFAME could be verified (Fig. 6d). A 5 nm hypsochromic shift of DFAME emission was detected when both fluorophores were present, pointing to electronic interaction between the two fluorophores bound to the heterodimeric RNA. This confirms that both fluorophore binding-sites of the heterodimer are being occupied. Importantly, particularly after considering the dissociation constants of the two RNAs from each ligand, it is likely that some of the heterodimers in solution are bound to two molecules of a same ligand. In those cases, the hypsochromic shift would not be present, suggesting that a solution with 100% of heterodimers being hetero-liganded may exhibit a more pronounced hypsochromic shift (Fig. 6d).

## Discussion

Comparison of the newly determined structures of Beetroot with those of the previously characterized Corn[17,18] reveals that despite their high sequence identity, these two aptamer RNAs have evolved fluorophore-binding sites in completely different locations (Fig. 1b, e). Both Beetroot and Corn are homodimeric RNAs, and their respective protomers fold as a duplex capped by a 4-tiered quadruplex. Beetroot homodimerizes by direct stacking of the exposed duplex-distal faces (T1 and t1) of the quadruplexes of each protomer (Fig. 2d). Previous analysis showed that the unliganded Corn RNA also dimerizes using the structurally analogous exposed quadruplex faces of its protomers[18]. However, the analogy does not extend to the fluorophore-bound states. In the case of DFAME-bound Beetroot, each protomer binds to one fluorophore molecule between the duplex-proximal face of its quadruplex (T4 and t4) and the adjacent duplex, resulting in a 2:2 RNA:fluorophore association that retains direct stacking of the quadruplexes of the two protomers. In contrast, the Corn homodimer binds one molecule of its cognate fluorophore, DFHO at its interprotomer interface[17], thereby interrupting the stacking of the quadruplexes of its two protomers to form a 2:1 RNA:fluorophore complex. Structure determination of Corn in complex with the non-specific, G-quadruplex-preferent fluorophore ThT revealed that this compound also occupies a binding site between the quadruplexes of Corn[18]. In marked contrast, Beetroot binds ThT (as well as DFHO) at the same locations where it binds DFAME, and with the same 2:2 RNA:fluorophore stoichiometry. ThT binding the same sites as DFAME (and DFHO) shows that the dimer interface of Beetroot is more resistant to invasion by the non-specific fluorophore than that of Corn. Despite its distinctly different binding sites in the two RNAs, ThT is strongly activated by both Beetroot and Corn. Overall, the stabilities interprotomer interfaces of Beetroot and Corn are distinctly different: the Beetroot interface remains unchanged in the face of specific or non-specific fluorophore binding, while the Corn interface is labile, allowing its two protomers to separate to bind fluorophores (Supplementary Fig. 11). The molecular basis for this marked difference remains to be elucidated.

Subtle sequence changes between the Beetroot and Corn quadruplexes (Fig. 1f) result in pronounced differences in their structures. The protomer Beetroot quadruplex consists of three canonical G-quartets and one mixed composition quartet, while that of Corn has two canonical G-quartets and two mixed composition quartets[17,18]. The structural differences between the two 4-tiered quadruplexes are most apparent when their respective connectivity is considered (Fig. 1g, h, Supplementary Fig. 3). In both cases, an all-parallel 2-tiered G-quadruplex (T1 and T2, both of which are canonical G-quartets) stacks on two additional quadruplex tiers of mixed connectivity, resulting in multiple inversions in the local direction of the RNA backbone between T2 and T3. In the simplest case, the four nucleotides that stack on each other in each of the four corners of the quadruplex would be contiguous in sequence (giving rise to the expected $G_4$-N-$G_4$-N-$G_4$-N-$G_4$ sequence motif that

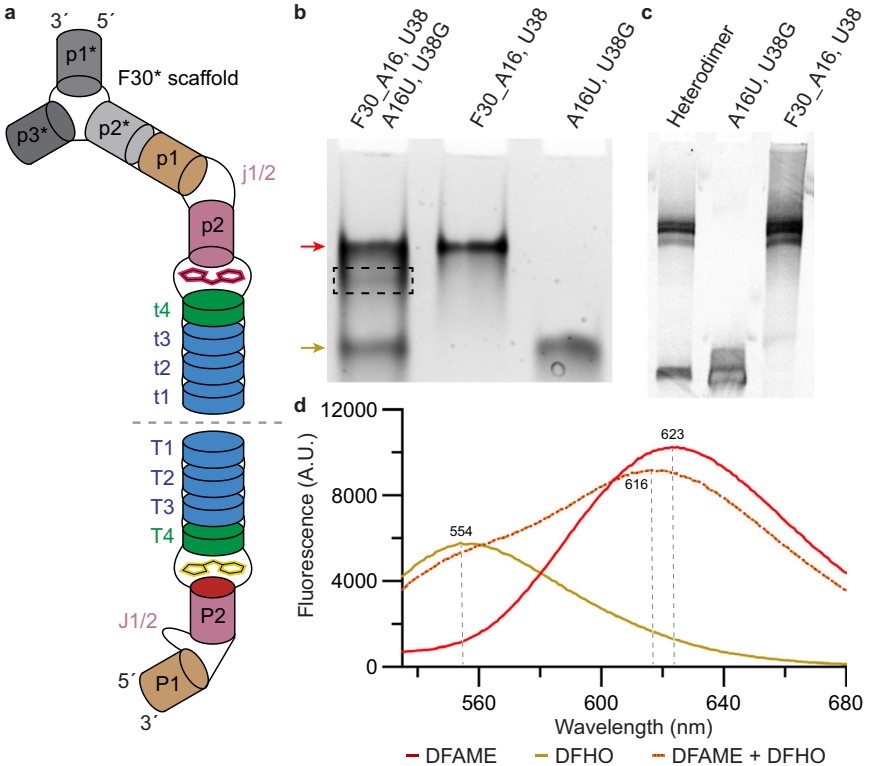

**Fig. 6 | The Beetroot fold supports heterodimer formation. a** Schematic depiction of the heterodimer formed by one wild-type protomer fused to F30 scaffold[23] and one Wobble Beetroot protomer. Asterisks denote secondary structure elements of the F30 scaffold. **b** Native polyacrylamide gel electrophoresis of the co-folded protomers and controls of F30 fused wild-type and Wobble homodimers. Dashed box indicates band excised for elution, red arrow indicates F30 fused wild-type homodimer, and mustard arrow indicates Wobble homodimer. Two independent experiments provided the same result. **c** Denaturing poly-acrylamide gel electrophoresis of the eluted RNA from **c** and controls F30 fused wild-type and Wobble RNAs. Two independent experiments provided the same result. **d** Emission spectra of the heterodimer in the presence of DFAME only (red), DFHO only (mustard), and DFAME + DFHO (red–mustard) when excited at 470 nm.

predicts a 4-tiered G-quadruplex, for instance) and chain direction inversions would be limited to unstacked, loop nucleotides. For the Beetroot quadruplex, this is not the case for any of the four nucleotide stacks, while it is true only for two of the stacks of Corn[17,18]. The elaborate connectivity of the Beetroot quadruplex allows four out of 16 nucleotides to be in the usually disfavored *syn*-glycosidic conformation[24–26], and seven out of its 16 riboses adopt the 2′-*endo* pucker that is normally associated with DNA (refs. 24,27). The unexpected differences between the quad-ruplexes of Beetroot and Corn underscore how even a high degree of sequence identity between two RNAs does not necessarily allow inference of a common folded structure or function.

The fluorophore-binding site of Beetroot is unusual among fluorogenic aptamer RNAs because of it relies exclusively on stacking interactions and shape complementarity to achieve selectivity. There are no RNA-fluorophore hydrogen bonds in any of the four Beetroot-fluorophore complex structures we have determined (Figs. 2, 3, 4). The structures of other RNAs that recognize GFP-derived fluorophores, such as Corn, Spinach and Squash (refs. 17,28,29) all show the use of direct or water-mediated hydrogen bonds as part of the recognition strategy. Comparison of the structures of Beetroot bound to DFAME, DFHO and ThT (Figs. 2, 3, 4) shows only small differences in the relative location of the fluorophores. Yet, change of a single base pair from A•U to U•G to generate Wobble Beetroot switched the fluorescence acti-vation selectivity of the aptamer from DFAME to DFHO by 12-fold. This is apparent not only in the fluorescence activation, but also in the similarity of fluorescence lifetimes between wild type–DFAME and Wobble Beetroot–DFHO, and in the changes in chemical shifts of the fluorophores as reported by [19]F NMR (Fig. 5). The structures of the two RNAs bound to DFHO differ in the pose of the two rings of the

fluorophore, and better interaction of its methyloxime substituent (Fig. 4d, e) demonstrating how small changes in RNA structure can have substantial functional effects. In this regard, it is noteworthy that the J1/2 element, which is functionally dispensable in Beetroot (and indeed, is partially disordered in our crystal structures) is immediately adjacent to the fluorophore binding site, and therefore represents an attractive location in which to engineer future Beetroot variants with improved or divergent fluorophore selectivity.

An unexpected finding of the initial biochemical characterization of Beetroot was that despite its sequence similarity to Corn, the two aptamers do not form mixed heterodimers[19]. The dimerization inter-faces of the two aptamers differ in that while Beetroot has one func-tionally important unpaired adenine that makes interprotomer interactions (Fig. 2d, e), Corn has three adenines that impart both, dimerization and fluorophore selectivity to that aptamer RNA (refs. 17,18). Despite the high stability of the Beetroot interprotomer interface, our calorimetric analysis (Fig. 2g, Supplementary Fig. 5) indicates that its two fluorophore binding sites are not thermo-dynamically linked. Nonetheless, as the two binding sites are separated by ~30 Å, their simultaneous occupancy by two fluorophores should result in through-space, electronic interaction. Our newly engineered variant, Wobble Beetroot, has a dimerization interface unchanged from that of its parental aptamer. Thus, unlike Corn heterodimers, which would activate a single fluorophore, Beetroot and Wobble Beetroot provided the so far unique opportunity to demonstrate for-mation of a heterodimer, that each protomer binds independently to activate the fluorescence of DFAME or DFHO, respectively, and that the two fluorophores interact electronically (Fig. 6). Our investigation of Beetroot represents the starting point for the discovery and develop-ment of new RNA functionalities, such as Förster resonance energy

transfer (FRET) systems based on fluorogenic aptamer dimerization and in RNA nanotechnology.

## Methods

### Fluorophores
3,5-difluoro-4-hydroxybenzylidene imidazolinone (DFHBI, **1**), (*Z*)−4-(3,5-Difluoro-4-hydroxybenzylidene)−1-methyl-5-oxo-4,5-dihydro-1*H*-imidazole-2-carbaldehyde oxime (DFHO, **2**), and 3,5-difluoro-4-hydroxybenzylidene imidazolinone-2-acrylate methyl (DFAME, **3**) and were synthesized as previously described[14,16,19] and used without further purification. Thioflavin T (ThT, **4**) was purchased from Sigma-Aldrich.

### RNA preparation
RNA constructs used in this study are listed in Supplementary Table 2. RNAs were transcribed from PCR templates with T7 RNA polymerase, and purified by denaturing gel electrophoresis (10% polyacrylamide, 29:1 acrylamide:bisacrylamide; 1 × TBE, 8 M urea). Where applicable, a 5′-hammerhead ribozyme served to generate 5′-OH termini[30]. After ultraviolet shadowing and excision from gels, RNAs were recovered by electroelution, washed with 1 M KCl, exchanged into 20 mM MOPS−KOH pH 7.0, 150 mM KCl, and 10 μM EDTA and concentrated by centrifugal ultrafiltration (Amicon Ultra, 10 kDa molecular weight cut-off, EMD Millipore), and stored at −20 °C. Prior to use, RNAs were heated to 85 °C for 3 min then allowed to cool at 21 °C for 10 min.

### Crystallization and diffraction data collection
RNAs were brought up to 1 mM MgCl₂, mixed with equimolar fluorophore (DFAME, DFHO, or ThT), then incubated at room temperature for 10 min. For crystallization by the sitting drop or hanging drop vapor diffusion methods, 0.2 μl of RNA-fluorophore solution (250 μM) and 0.2 μl reservoir solution [2.0 M ammonium sulfate; 5% (v/v) 2-propanol for Beetroot and 2.4 M ammonium sulfate, 5% (v/v) 2-propanol for Beetroot A16U,U38G] were mixed and equilibrated at 21 °C. Strongly fluorescent (500 nm illumination, Supplementary Figure 1), plate-shaped crystals grew to maximum dimensions of 150 × 75 × 10 μm³ over 3−10 days. Two min after addition of 0.5 μl of a solution comprising 10 mM MOPS−KOH pH 7.0, 75 mM KCl, 5 μM EDTA, 500 μM MgCl₂, 2.5 M Na malonate, 125 μM fluorophore, and 25% (v/v) glycerol to the drops, crystals were mounted in nylon loops and flash-frozen by plunging into liquid nitrogen. Diffraction data (Supplementary Table 1) were collected at 100 K using the rotation method at beamlines 5.0.1 or 5.0.2 of the Advanced Light Source (ALS), or beamlines 24-ID-C or 24-ID-E of the Advanced Photon Source (APS), or 24-ID-D (CCP4-APS Summer School) and reduced using xia2 (ref. 31) with Dials (ref. 32) (Beetroot-DFAME, Beetroot-ThT, and Beetroot A16U,U38G-DFHO complexes) or XDS (ref. 33) (Beetroot-DFHO complex).

### Structure determination and refinement
The structure of the Beetroot-DFAME complex was solved by molecular replacement using Phaser-MR (ref. 34) and a search model consisting of 2 copies of the 2-tiered G-quadruplex of the Corn structure[17] (PDB: 5BJO). The best solution (log-likelihood gain of 521 and translation function Z-score of 19.3) was subjected to manual rebuilding in Coot (ref. 35), interspersed with rounds of simulated annealing, energy minimization and individual B-factor refinement in Phenix (ref. 36) resulting in a model with $R_{free}$ = 0.204 at 1.95-Å resolution. To determine the orientation of the fluorophore, it was built in either of the two possible poses, and the model refined. The incorrect orientation exhibited a - 5 σ peak in the resulting $|F_o|$-$|F_c|$ Fourier syntheses. The structure of the Beetroot-DFHO, Beetroot-ThT, and Beetroot A16U,U38G-DFHO complexes were also solved by molecular replacement[34] using a search model consisting of the RNA atoms (without P1, J1/2, and P2) of the refined Beetroot-DFAME structure. Solutions were subjected to rounds of simulated annealing, energy minimization and B-factor refinement[36] interspersed with manual

rebuilding[35]. The correct orientation of fluorophores, where ambiguous in $2|F_o|$-$|F_c|$ maps, was determined as for the DFAME complex. For all structures, the mean precision of atomic coordinates was estimated using Phenix (ref. 36). Refinement statistics are summarized in Supplementary Table 1. In the Beetroot-DFHO and Beetroot-ThT structures, due to poor electron density in some regions (P1 and J1/2), some nucleotides were modeled but refined with occupancy=0 (Beetroot-DFHO: G1, C2, G48, C49, g6, c44, c47, and c49; Beetroot-ThT = C2, G48, C49, g3, c4, g6, u9, g46, c47, g48, and c49). Structural figures and r.m.s.d. calculations were prepared with PyMOL (ref. 37). Solvent-accessible surface areas were calculated using AREAIMOL − CCP4 (ref. 38).

### Fluorescence spectroscopy
Fluorescence scans were performed on a Photon Technologies International/820 Photomultiplier Detection System or on a CLARIOstar Plus 0430 (BMG Labtech) set to excite and measure emission at wavelengths of interest. After preparation and brought up to 5 mM Mg²⁺, RNAs were mixed with at least equimolar ligand concentrations. DFHBI was excited at 450 nm and emission measured at 530 nm. DFHO was excited at 470 nm and emission measured at 560 nm. DFAME was excited at 474 nm and emission measured at 625 nm. ThT was excited at 455 nm and emission measured at 485 nm. DFHO + DFAME mix was excited at 470 nm and emission measured at 535−680 nm range.

### Fluorescence lifetime measurements
Lifetime measurements were performed using a lab-built time-correlated single photon counting system (TCSPC). This system consists of a diode-pumped solid-state laser (Verdi-V10, Coherent) that pumps a Ti:sapphire laser (Mira 900-D, Coherent) operating in fs mode with a repetition rate of 76 MHz. The Ti:sapphire was tuned to central wavelengths of 900 nm and 940 nm. The resultant near-IR light was then pulse picked with an electro-optic modulator (Model 350-160-02, ConOptics) and frequency doubled with a BBO crystal to produce the excitation sources of 450 nm and 470 nm. Samples were placed in FireflySci 10 × 2 mm dual pathlength UV quartz cuvettes and excited with vertically polarized light through the 2 mm path. The resulting emission was filtered to remove the excitation source and collected at the magic angle using a JYH10 monochromator with an 8 nm bandwidth and a water cooled MCP photomultiplier. The instrument response function (IRF), typically <150 ps, was recorded daily at the excitation wavelengths from a light-scattering suspension of dilute colloidal silica. A standard solution of rhodamine 6 g in ethanol was taken daily and used to determine color shift (if any) of the detector. Fluorescence lifetimes were determined by fitting the fluorescence decay curves using our in-house program DecayFit, which uses a least squares algorithm that reconvolutes the measured IRF with the decay before fitting. The number of exponentials (*n* in Eq. 1) used to fit the data was determined by optimizing χ2. The statistical relevance of added sequential fit parameters was verified with an F-test. For fluorescence decays of the same sample over multiple emission wavelengths (10 nm spacing) our lab-built software tfitz was used. This also uses a least square algorithm, but allows for a global linkage of lifetimes over multiple decay curves.

$$I(t) = \sum_{i=1}^{n} a_i e^{-\frac{t}{\tau_i}} \quad (1)$$

### Native purification of heterodimers
F30[23] scaffold was fused to Beetroot so it would present a considerably different length than Beetroot variant A16U,U38G (Supplementary Table 2). Equimolar amounts of Beetroot-F30 scaffold[23] and Beetroot A16U,U38G were pre-mixed and folded as previously described. RNAs were brought up to 5 mM MgCl₂ and equimolar ligand of interest

(DFAME, DFHO, or both DFAME and DFHO) was added to the mix. The mixture was than purified in 6% non-denaturing PAGE with 1× TBE buffer containing 50 mM KCl and 5 mM MgCl₂. Band of interest was either visualized by EtBr dye or ultraviolet shadowing and excised from gels. RNAs were eluted from gel by incubation with 20 mM MOPS-KOH pH 7, 150 mM KCl and 5 mM MgCl₂ and 10 µM EDTA. Controls F30-Beetroot and Beetroot 16U-U38G were used as molecular weight markers (34717 $g/mol$ and 15239 $g/mol$, respectively). When needed, RNAs were concentrated by gentle centrifugal ultrafiltration (Amicon Ultra, 10 kDa molecular weight cut-off). The heterodimer stability was resolved (48 h after purification) in 6% non-denaturing PAGE with 1× TBE buffer containing 50 mM KCl and 5 mM MgCl₂. Uncropped gels are provided in the Source Data file.

## NMR spectroscopy

A 250 µl solution of 450 µM Beetroot RNA, variant Beetroot RNA A16U,U38G, or hammerhead ribozyme, 500 µM DFAME or 500 µM DFHO in 20 mM MOPS-KOH pH 7, 75 mM KCl, and 5 mM MgCl2 was prepared and folded as previously described. Samples were transferred to a Shigemi tube and supplemented with 8% D2O to provide a lock signal. ¹⁹F NMR spectra were recorded at 308 K with a 5 mm QXI ²H/¹H/¹⁹F-¹³C/¹⁵N XYZ-gradient probe on a 600 MHz Bruker Avance III NMR system operating at a ¹⁹F Larmor frequency of 565.08 MHz with the carrier placed at −135.00 ppm. Spectra were referenced to the fluorine resonance of trifluoroacetic acid at −75.5 ppm, prepared as an external reference of 10% TFA in buffer containing 8% D2O. One-dimensional ¹⁹F spectra were recorded with a spectral window of 10.0 ppm, 4096 complex points, 2048 scans and a recycle delay of 1 s. All spectra were processed and analyzed using TopSpin v.3.0 (Bruker). Error estimates for ¹⁹F chemical shift were performed by acquiring 1D spectra in triplicate at different times and determining the standard deviation for one of the sharper free ligand resonances ( + 0.001 ppm) and for one of the broader, loosely bound ligand resonances ( + 0.002 ppm).

## Isothermal titration calorimetry

ITC measurements were performed using a MicroCal iTC200 micro-calorimeter (GE) at 25 °C using 16 injections and 180 s delays. A solution of 300 µM of fluorophore DFAME was titrated into a 30 µM sample of folded RNA with 5 mM MgCl₂. Data were analyzed and fit using NITPIC (ref. 39) and SEDPHAT (ref. 40).

## Size-exclusion chromatography

RNAs were analyzed on a Superdex 75 Increase (24 mL bed volume) size-exclusion column (GE Life Sciences) at 0.75 ml/min at 21 °C using a mobile phase composed of 20 mM Mops (pH 7.0), 150 mM KCl, 10 µM EDTA, and 5 mM MgCl₂. Absorbance was monitored at 260, 280, and 295 nm.

## Size-exclusion chromatography coupled to multi angle light scattering

SEC-MALS was performed on a Dawn HELEOS-II MALS with QELS DSL and an Optilab T-rEX differential refractive index detector (Wyatt Technology) at 0.5 ml/min over a SW3000 column (Tosoh Bioscience) at room temperature using same buffer as SEC with the addition of 0.05% sodium azide. Data were analyzed in Astra 7 (Wyatt Technology, CA, USA).

## Reporting summary

Further information on research design is available in the Nature Portfolio Reporting Summary linked to this article.

## Data availability

The atomic coordinates and structure factor amplitudes have been deposited with Protein Data bank under accession codes 8EYU (Beetroot–DFAME), 8EYV (Beetroot–DFHO), 8EYW (Beetroot–ThT), and 8F0N (Beetroot A16U,U38G–DFHO). The RNA aptamer Corn data used in this study is available in the Protein Data bank database under accession code 5BJO. The fluorescence, uncropped gels, pictures of the plate-shaped Beetroot–DFAME co-crystals, size-exclusion chromatography, and isothermal titration calorimetry data generated in this study are provided in the Source Data file. Any additional data required will be made available upon request. Source data are provided with this paper.

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

## Acknowledgements

We thank the staff of beamlines 5.0.1 and 5.0.2 of the Advanced Light Source, Lawrence Berkeley National Laboratory (ALS), 24-ID-C and 24-ID-E of the Advanced Photon Source, Argonne National Laboratory (APS) and 22-ID-D (CCP4-APS Summer School) for crystallographic data collection; G. Piszczek and D. Wu of the Biophysics Core of the National Heart, Lung and Blood Institute (NHLBI) for fluorescence, ITC, and SEC-MALS; and M. Banco, N. Demeshkina, A. Elghondakly, C. Jones, and R. Trachman for discussions. This research used resources of the APS, a U.S. Department of Energy (DOE) Office of Science User Facility operated for the DOE Office of Science by Argonne National Laboratory under Contract No. DE-AC02-06CH11357. This work is based upon research conducted at the Northeastern Collaborative Access Team beamlines, which are funded by the National Institute of General Medical Sciences from the National Institutes of Health (NIH P30 GM124165). The Pilatus 6M detector on 24-ID-C beam line is funded by a NIH-ORIP HEI grant (S10 RR029205). This work was supported in part by NIH grant R35NS111631 (to S.R.J.) and by the intramural program of the NHLBI, NIH.

## Author contributions

S.R.J. and A.R.F. initiated the project; L.F.M.P. performed fluorescence, crystallographic, and ITC experiments; M.R.S. and N.T. performed NMR, K.A.L. and J.K. performed fluorescence lifetime measurements, J.W. performed initial fluorescence and biochemical measurements, A.R.F. and L.F.M.P. wrote the manuscript with contributions from all authors.

## Funding

## Competing interests

S.R.J. is the co-founder and has equity in Chimerna Therapeutics and Lucerna Technologies. Lucerna has licensed technology related to Spinach and other RNA-fluorophore complexes. All other authors declare no competing interest.
