## [Peer Review File · Nature Communications]

Co-crystal structures of the fluorogenic aptamer Beetroot show that close homology may not predict similar RNA architecture.REVIEWER COMMENTS

Reviewer #1 (Remarks to the Author):

The manuscript by Passalacqua and coauthors describes a structure of the fluorogenic RNA called Beetroot. Despite sequence similarity with the “Corn” fluorogenic RNA, Beetroot RNA forms a different three-dimensional structure and uses different features for recognition of the fluorogenic ligands. This study will be of great interest to the RNA researchers as well as many biologists since fluorogenic RNAs open an avenue to study behaviors of RNAs in vitro and live cells.

The study conducted and written well. I do not see any major problem. Minor issues and suggestions are listed below.

- 1) According to the original definition, a protomer is the structural unit of an oligomeric protein. Therefore, it is incorrect to call RNA monomers “protomers”.
- 2) Line 96. DFAME (3) not 2.
- 3) Line 146. Watson-Crick•Sugar edge interactions not Watson-Crick•Hoogsteen interactions. See points 5 and 6 to increase clarity.
- 4) Line 146-147, no hydrogen bonds are present between the bases of A34 and A37 not G17. See points 5 and 6 to increase clarity.
- 5) Line 149. Stacking interactions are important for the recognition of the fluorophores and are critical to understand the system. I strongly recommended to move ED Fig. 4 to the main text Fig. 2. Also, it would be beneficial to have panels d, e, and f of Fig. 4 next to the corresponding side views in Figs. 2, 3 and 4 to provide
- 6) Line 221-223. It is awkward to look at Fig. 4e and f before panels a-d. As explained in point 2, moving top views to the corresponding figures will help to better understand the ligand recognition. I am not sure that it is best to have the top views together after presenting the corresponding structures.
- 7) In order to directly compare Wobble Beetroot mutant with other RNA(s), I recommend to prepare an additional panel. This panel can show superposition of panel 4g for Wobble Beetroot-DFHO (in colors, without G38-U16) with, for example, panel e for Beetroot-DFHO (in gray). Having superposed ligands would make comparison easier to understand.
- 8) Line 159. Fig 1c not Fig1c-e.
- 9) ΔH is 3.58 ± 0.13 kcal/Mol in ED Fig. 6. and -3.58 ± 0.13 kcal/Mol line 192.
- 10) Line 264-265. “Downfield chemical shifts changes indicate that the fluorophores are in a more electron-withdrawing environment”. Could this be associated with the hydrophobic effect? In both instances, fluorinated phenyl rings are sandwiched between either A16 and G17 or U16 and G17.
- 11) Fig. 5. and line 260-264. The NMR peak shifts are incredibly small (as low as 0.02 ppm). How reliable are these measurements, especially considering peak broadening? Is this because of DFAME or DFHO forming non-specific interactions with Beetroot? In the Squash paper, the shifts were 10-fold bigger (0.5 ppm). To avoid confusion and clarify the issue, the authors should perform the ^{19}F NMR measurements with DFAME or DFHO and RNAs uncappable of specific binding to DFAME or DFHO molecules under the same conditions.
- 12) Please use the same abbreviations and designations throughout the manuscript. For example, use either Kd or KD.

- 13) Please update the structures to decrease the number of ligand bond length and angle outliers as indicated by a large number of outliers in the validation reports. I believe that .cif files have to be updated according to restraints used in Mogul.
- 14) Please double-check the structures. For example, waters 124, 131, 179, etc. in the DFAME structure are not in place. A number of water molecules have weak maps, sometimes emerging at 0.7 sigma level.
- 15) I am not sure that the .pdb files supplied by the authors are the same files used for PDB validation.

Reviewer #2 (Remarks to the Author):

The authors report several co-crystal structures of a recently published fluorescent aptamer-dye system, beetroot/DFAME, and complement them with mutational, binding, fluorescence, and NMR data. They uncover key differences compared to the related aptamer corn and compare the binding of different dyes. In addition, they generate a mutant "wobble" variant that they claim is orthogonal to the "wild type" of beetroot. For this variant, they claim "a 12-fold selectivity switch." Finally, they are developing a heterodimeric version that is claimed to be an RNA analog to split-GFP.

Overall, this work is of great importance to the narrower field of RNA imaging systems and the broader field of RNA structural biology, as it shows how subtle differences in RNA sequence and overall folding can lead to completely different ligand binding abilities.

1. I disagree with the authors' use of the term "orthogonal." The manuscript lists the following binding dissociation constants K_D (on pages 10 and 13), all determined by fluorescence spectroscopy:

- Beetroot - DFAME 1.55 μM ;
- Beetroot - DFHO 9 μM ;
- Wobble - DFAME 5 μM ;
- Wobble - DFHO 7.5 μM .

These data show that both aptamers prefer DFAME over DFHO, implying that they are not orthogonal.

2. The authors postulate (even in the abstract) a 12-fold selectivity switch between the two aptamers. While there are various measures of selectivity, by far the most common is based on a quantitative comparison of k_D values. The k_D values described above give a selectivity ratio of 3.9, not 12, and I do not think it is appropriate to postulate selectivity from uncorrected fluorescence light-up factors.

3. no standard deviations are given for the k_D values.

4. since all k_D values are very similar (within a factor of six), any claims that the heterodimeric aptamer would be hetero-liganded are speculative. Based on the k_D values, one would expect that the most abundant species would be the aptamer dimer bound to two molecules of DFAME. In my opinion, the evidence presented (Fig. 6d) is rather weak. Moreover, it is not clear from the description of the experiment exactly how the authors did this. How did they ensure that no dissociation of the heterodimer occurred during the experiment?

5. the analogy of the heterodimeric system to split-GFP is a stretch. In split-GFP, only the fully assembled protein can fluoresce, whereas in the heterodimeric aptamer, each half can fluoresce independently of the other.

6. Summary: "...discover that this RNA binds two independent molecules of the fluorophore" I suggest

adding the word "dimer" to RNA, as the reader might understand this as an aptamer binding two molecules of the ligand. The same applies to the end of the introduction (p.5).

7. p.9, middle: "First, mutation of any of the guanines in the G-quadruplex abolished fluorescence (not shown)." Really? Since there are 13 guanines in the G-quadruplex, there would be 39 different possible single mutations that do not appear in the supplemental table of sequences. The authors should be more specific about which mutants were tested.

Reviewer #3 (Remarks to the Author):

The manuscript describes the structural analysis of a fluorescence turn-on aptamer that binds a organic heteroaromatic molecule. RNA-binding increases its fluorescent properties. The in-vitro selected molecules are important as in-cell probe for RNA localisation, transcription yields and monitoring coupled transcription-translation in cell-free assays.

The work presented here is thus timely, and the structural and biophysical data on such systems are scarce. Any additional information of the quality presented here is highly appreciated.

The particular aspect presented here is the newly identified turn-on aptamer Beetroot that exhibits (i) a high dimerization tendency and (ii) binds two molecules of heteroaromatic molecules, one per monomer. This is different to CORN, a previously engineered fluorescence turn-on RNA aptamer. The fact that Beetroot binds two molecules, 30 Angstrom apart, and might be able to form heterodimeric complexes allows also for more sophisticated advanced light microscopy experiments. For their development and optimization, the structural data presented here will form an important starting points.

The data are very interested and very well presented. I thus recommend publication of the data with minor changes.

1.) minor comment: in abstract and main text: do not state two "independent" molecules, but simple two molecules (p.2, line 32, p.5, line 85).

2.) If I understand correctly, than ITC data show that binding is both, enthalpically and entropically favorable. Thus, rephrase on page 10, line 191 to state. "that the association is MAINLY entropically driven,..."

3.) P. 13 describes the NMR analysis of binding of the heteroaromatic molecules to the RNA. I am slightly puzzled about the statements made and would rephrase those: Firstly, the downfield chemical shift changes may or may not indicate a more electron-withdrawing environment. I recommend removing this interpretation. "...suggesting more restricted motion for DFHO..." - I am not convinced that this is corrected. In fact, chemical shift modulation within the binding pocket of DFHO would be more convincing. The authors have not conducted the appropriate NMR experiments (T2 vs. T1rho experiments) to convincingly document their assumption. I recommend removing this interpretation.

4.) Figure 1a chemical structure -> chemical structures.

We thank the reviewers for their insightful comments. Our responses below are in Italics.

Reviewer #1 (Remarks to the Author):

The manuscript by Passalacqua and coauthors describes a structure of the fluorogenic RNA called Beetroot. Despite sequence similarity with the "Corn" fluorogenic RNA, Beetroot RNA forms a different three-dimensional structure and uses different features for recognition of the fluorogenic ligands. This study will be of great interest to the RNA researchers as well as many biologists since fluorogenic RNAs open an avenue to study behaviors of RNAs in vitro and live cells. The study conducted and written well. I do not see any major problem. Minor issues and suggestions are listed below.

1) According to the original definition, a protomer is the structural unit of an oligomeric protein. Therefore, it is incorrect to call RNA monomers "protomers".

Reply: We can see the perspective of the reviewer. However, we note that much of the terminology for describing complex RNA structure is derived from that developed originally to describe proteins. Thus, it is accepted practice to describe long-range RNA structural interactions as "tertiary contacts" and the fold of complex RNAs as "tertiary structure", even though that terminology was developed originally to describe protein structure. By the same token, it is accepted practice to use the term "secondary structure" to describe the pattern of double helices in an RNA, even though no RNA has alpha helices or beta strands, and unlike protein secondary structure elements, which are strictly local, all RNA duplexes involve close contact of non-sequence-adjacent segments of the polymer. Our point is that terminology developed for proteins can be (and has been, by many) extended by analogy (mutatis mutandis) to RNA when this is the most economical way to describe a structural feature. In the case of Beetroot (and Corn, previously) we think that the term "monomer" should be reserved for a folded single RNA chain that exists independently in solution, and "dimer" to the association of two chains. Then, to describe "half of the homodimer", we need to use a different term. We think "protomer" is appropriate, rather than "subunit". In common usage, "protomer" describes similar or identical folded and functional protein chains that form an oligomer (e.g., the protomers of the "rotor" of ATP-synthetase), whereas "subunit" is often used to describe different protein chains forming a higher-order complex (e.g., the alpha and beta subunits of RNA polymerase), but we are open to replacing "protomer" throughout with "subunit" if the referee thinks this would improve clarity.

2) Line 96. DFAME (3) not 2.

Reply: Thank you for noticing this error. We corrected it accordingly.

3) Line 146. Watson-Crick•Sugar edge interactions not Watson-Crick•Hoogsteen interactions. See points 5 and 6 to increase clarity.

Reply: Thank you for noticing this. We have corrected accordingly.

4) Line 146-147, no hydrogen bonds are present between the bases of A34 and A37 not G17. See points 5 and 6 to increase clarity.

Reply: Thank you for noticing this. We have corrected accordingly.

5) Line 149. Stacking interactions are important for the recognition of the fluorophores and are critical to understand the system. I strongly recommended to move ED Fig. 4 to the main text Fig. 2. Also, it would be beneficial to have panels d, e, and f of Fig. 4 next to the corresponding side views in Figs. 2, 3 and 4 to provide ...

Reply: We appreciate the suggestions, and we followed them accordingly. Indeed, reorganizing the figures as suggested has increased the clarity of the manuscript.

6) Line 221-223. It is awkward to look at Fig. 4e and f before panels a-d. As explained in point 2, moving top views to the corresponding figures will help to better understand the ligand recognition. I am not sure that it is best to have the top views together after presenting the corresponding structures.

Reply: As in item 5, we appreciate the suggestion and followed them accordingly.

7) In order to directly compare Wobble Beetroot mutant with other RNA(s), I recommend to prepare an additional panel. This panel can show superposition of panel 4g for Wobble Beetroot-DFHO (in colors, without G38-U16) with, for example, panel e for Beetroot-DFHO (in gray). Having superposed ligands would make comparison easier to understand.

Reply: As in item 5 and 6, we appreciate the suggestions and followed them accordingly.

8) Line 159. Fig 1c not Fig1c-e.

Reply: Thank you for noticing. We corrected accordingly.

9) ΔH is 3.58 ± 0.13 kcal/Mol in ED Fig. 6. and -3.58 ± 0.13 kcal/Mol line 192.

Reply: Thank you for noticing. We corrected the Extended Data Fig. 6 (now Extended Data Fig. 5) accordingly.

10) Line 264-265. "Downfield chemical shifts changes indicate that the fluorophores are in a more electron-withdrawing environment". Could this be associated with the hydrophobic effect? In both instances, fluorinated phenyl rings are sandwiched between either A16 and G17 or U16 and G17.

Reply: We appreciate the comment by the reviewer. We agree that the hydrophobic effect may also play a role in the shifts we see. This comment is related to comment # 3 of reviewer 3 and we modified the text to take both suggestions into account.

11) Fig. 5. and line 260-264. The NMR peak shifts are incredibly small (as low as 0.02 ppm). How reliable are these measurements, especially considering peak broadening? Is this because of DFAME or DFHO forming non-specific interactions with Beetroot? In the Squash paper, the shifts were 10-fold bigger (0.5 ppm). To avoid confusion and clarify the issue, the authors should perform the ^{19}F NMR measurements with DFAME or DFHO and RNAs uncappable of specific binding to DFAME or DFHO molecules under the same conditions.

Reply: We thank the reviewer for the comment and experiment suggestion. We evaluated the experimental error in our chemical shifts by carrying out the experiment at three random times. The reproducibility is one magnitude smaller (0.001-0.002ppm) than the differences we are reporting here. This error is in line with the estimated error based on our S/N and linewidths of the resonances as published recently (Chilivery et. al., Chem. Rev. 2022, 9307-9330). Indeed, the peak shift is very small and should be analyzed

carefully. One of the reasons we think that the shift is not very pronounced and broadening happens is due to the openness of the binding site of Beetroot. The Beetroot RNA buries ~ 40% of the ligands tested here, a modest value when compared to 87% as reported to Squash–DFHBI-1T. We did perform the experiment suggested using the cleaved hammerhead ribozyme (HHR) served to generate 5'-OH termini Beetroot. Interestingly, the NMR results of DFAME + HHR showed a similar profile to the DFAME + Wobble Beetroot non-cognate pair, with a +0.02 ppm shift, suggesting non-specific interaction(s). To confirm that this non-specific interaction does not yield fluorescence, we tested it for fluorescence activation. The results showed that no fluorescence activation happens when DFAME is excited in the presence of HHR. We added a new figure with these results (Extended Data Fig. 7). We also comment the results in the manuscript.

12) Please use the same abbreviations and designations throughout the manuscript. For example, use either *K_d* or *K_D*.

Reply: We have checked the manuscript for consistency.

13) Please update the structures to decrease the number of ligand bond length and angle outliers as indicated by a large number of outliers in the validation reports. I believe that .cif files have to be updated according to restraints used in Mogul.

Reply: We appreciate the suggestion. We have made the changes accordingly.

14) Please double-check the structures. For example, waters 124, 131, 179, etc. in the DFAME structure are not in place. A number of water molecules have weak maps, sometimes emerging at 0.7 sigma level.

Reply: We appreciate the suggestions. We have made the changes accordingly.

15) I am not sure that the .pdb files supplied by the authors are the same files used for PDB validation.

Reply: We sent to the reviewers the same .pdb files that we used for the PDB validation.

Reviewer #2 (Remarks to the Author):

The authors report several co-crystal structures of a recently published fluorescent aptamer-dye system, beetroot/DFAME, and complement them with mutational, binding, fluorescence, and NMR data. They uncover key differences compared to the related aptamer corn and compare the binding of different dyes. In addition, they generate a mutant "wobble" variant that they claim is orthogonal to the "wild type" of beetroot. For this variant, they claim "a 12-fold selectivity switch." Finally, they are developing a heterodimeric version that is claimed to be an RNA analog to split-GFP. Overall, this work is of great importance to the narrower field of RNA imaging systems and the broader field of RNA structural biology, as it shows how subtle differences in RNA sequence and overall folding can lead to completely different ligand binding abilities.

1. I disagree with the authors' use of the term "orthogonal." The manuscript lists the following binding dissociation constants *K_D* (on pages 10 and 13), all determined by fluorescence spectroscopy:

- Beetroot - DFAME 1.55 μ M;
- Beetroot - DFHO 9 μ M;
- Wobble - DFAME 5 μ M;

-Wobble - DFHO 7.5 μ M.

These data show that both aptamers prefer DFAME over DFHO, implying that they are not orthogonal.

Reply: We thank the reviewer for raising this point. We meant orthogonality in the fluorescence activation and not binding (we do mention in the manuscript that fluorescence activation does not correlate directly with binding affinity). We have changed the wording accordingly to make it clear we mean fluorescence activation and not binding.

2. The authors postulate (even in the abstract) a 12-fold selectivity switch between the two aptamers. While there are various measures of selectivity, by far the most common is based on a quantitative comparison of K_D values. The K_D values described above give a selectivity ratio of 3.9, not 12, and I do not think it is appropriate to postulate selectivity from uncorrected fluorescence light-up factors.

Reply: As previously in number 1, we indeed failed to specify that we meant fluorescence activation and thus it needs to be specifically addressed like that. We thank the reviewer, and we made the changes accordingly.

3. no standard deviations are given for the K_D values.

Reply: We apologize for not including them in the main text. The values with standard deviations were included in each respective figure legend.

4. since all K_D values are very similar (within a factor of six), any claims that the heterodimeric aptamer would be hetero-liganded are speculative. Based on the K_D values, one would expect that the most abundant species would be the aptamer dimer bound to two molecules of DFAME. In my opinion, the evidence presented (Fig. 6d) is rather weak. Moreover, it is not clear from the description of the experiment exactly how the authors did this. How did they ensure that no dissociation of the heterodimer occurred during the experiment?

Reply: We appreciate the comments. We agree that is likely that some heterodimers will be bound to two molecules of the same ligand in some cases and in others will be hetero-liganded. However, based on our controls using single ligands added to the data that shows that the Wobble Beetroot activates poorly DFAME fluorescence and the hypsochromic shift indicates that hetero-liganded heterodimers are formed in solution. We do agree that discussion as suggested by the reviewer is needed, and we added it to the manuscript. Regarding the dissociation control, we followed the heterodimer stability up to 48 hours and we did not see any indication of dissociation and homodimer formation. We added a figure of the 48 hours gel to Extended Data Fig. 10.

5. the analogy of the heterodimeric system to split-GFP is a stretch. In split-GFP, only the fully assembled protein can fluoresce, whereas in the heterodimeric aptamer, each half can fluoresce independently of the other.

Reply: Thus far, there is no evidence that a monomeric Beetroot can activate a fluorophore. Indeed, it is not clear that a monomer can exist stably folded in solution. As this remains to be established, we have removed references to split GFP accordingly.

6. Summary: "...discover that this RNA binds two independent molecules of the fluorophore" I suggest adding the word "dimer" to RNA, as the reader might understand this as an aptamer binding two molecules of the ligand. The same applies to the end of the introduction(p.5).

Reply: Thank you for the suggestion. We have added the word "dimer" in both instances.

7. p.9, middle: "First, mutation of any of the guanines in the G-quadruplex abolished fluorescence (not shown)." Really? Since there are 13 guanines in the G-quadruplex, there would be 39 different possible single mutations that do not appear in the supplemental table of sequences. The authors should be more specific about which mutants were tested.

Reply: We thank the reviewer for this comment. We did not express the correct message we wanted, and it is confusing and misleading the way it's written. We meant that any of the guanines that we mutated (that are not the 39 possibilities) did not yield any active aptamer. We mainly tried to add more Gs to the mixed tetrad T4 to verify if this could make the aptamer more stable and tried to disrupt T1 G-quartet to verify if the aptamer could be active without the top tier and work as a monomer. We have fixed the text and added the list of mutations we did characterize in the supplemental table.

Reviewer #3 (Remarks to the Author):

The manuscript describes the structural analysis of a fluorescence turn-on aptamer that binds an organic heteroaromatic molecule. RNA-binding increases its fluorescent properties. The in-vitro selected molecules are important as in-cell probe for RNA localization, transcription yields and monitoring coupled transcription-translation in cell-free assays. The work presented here is thus timely, and the structural and biophysical data on such systems are scarce. Any additional information of the quality presented here is highly appreciated. The particular aspect presented here is the newly identified turn-on aptamer Beetroot that exhibits (i) a high dimerization tendency and (ii) binds two molecules of heteroaromatic molecules, one per monomer. This is different to CORN, a previously engineered fluorescence turn-on RNA aptamer. The fact that Beetroot binds two molecules, 30 Angstrom apart, and might be able to form heterodimeric complexes allows also for more sophisticated advanced light microscopy experiments. For their development and optimization, the structural data presented here will form an important starting points. The data are very interested and very well presented. I thus recommend publication of the data with minor changes.

1.) minor comment: in abstract and main text: do not state two "independent" molecules, but simple two molecules (p.2, line 32, p.5, line 85).

Reply: Thank you for the suggestion. We corrected it accordingly.

2.) If I understand correctly, than ITC data show that binding is both, enthalpically and entropically favorable. Thus, rephrase on page 10, line 191 to state. "that the association is MAINLY entropically driven,..."

Reply: We appreciate the suggestion. We corrected it accordingly.

3.) P. 13 describes the NMR analysis of binding of the heteroaromatic molecules to the RNA. I am slightly puzzled about the statements made and would rephrase those: Firstly, the downfield chemical shift changes may or may not indicate a more electron-withdrawing environment. I recommend removing this interpretation. "...suggesting more restricted motion for DFHO..." - I am not convinced that this is corrected. In fact, chemical shift modulation within the binding pocket of DFHO would be more convincing. The authors have not conducted the appropriate NMR experiments (T2 vs. T1rho experiments) to convincingly document their assumption. I recommend removing this interpretation.

Reply: We thank the reviewer for the comment. We have changed the text accordingly. We agree with this reviewer completely. The ¹⁹F chemical shift is quite sensitive as

illustrated by the free DFAME and DFHO showing ^{19}F chemical shift difference of 0.1 ppm due to modification at the imidazole ring which is quite distant from the fluoro-phenyl ring. If the ligands had been tightly bound, the stacking of the RNA bases above and below the fluorophore should have produced significant ^{19}F shift perturbation. We suspect that the openness of the binding site allows for chemical shift modulation (due to the dynamic) of the fluorophore in the bound state, resulting in very small observed chemical shift differences.

4.) Figure 1a chemical structure -> chemical structures.

Reply: Thank you for noticing this. We corrected the figure legend accordingly.

REVIEWERS' COMMENTS

Reviewer #1 (Remarks to the Author):

My comments are attached as a separate file.

The revised manuscript has been significantly improved and the authors addressed the majority of my inquiries. I have only two comments (in blue).

1)

I leave it to the authors to decide on the terminology to describe their structures, although I do not object calling a half of a dimer a monomer, especially since each half of the RNA dimer can have its own “function”, binding to distinct fluorophores in the context of a “heterodimer”. In the protein field, researchers often use the term “monomer” to describe halves of dimers even if each monomer does not have, for example, enzymatic activity on its own and requires extensive contacts and rearrangements in the dimeric state to have catalytic activity.

13)

It is difficult to see the appropriate changes in all structures because only a single validation report has been submitted with the revised manuscript. However, in the submitted report there are no improvements in the quality of the ligand structure. For example, new validation report lists 21 bond length outliers, the same number as before. It is over 40% of all bonds. While it could be more challenging to fix angle and torsion angle outliers and the library values could be disputed, the bond outliers are clearly bad outliers with values deviating significantly (~10%) from the ideal values. The authors may want to fix these outliers.

Reviewer #2 (Remarks to the Author):

The authors have addressed all points raised by me in a constructive manner. I am confident that the quality and clarity of the manuscript has been significantly improved through the revision. I recommend the manuscript for publication.

Reviewer #3 (Remarks to the Author):

The manuscript can now be accepted for publication.

We thank the reviewers for their insightful comments. Our responses below are in Italics.

Reviewer #1 (Remarks to the Author):

My comments are attached as a separate file.

The revised manuscript has been significantly improved and the authors addressed the majority of my inquiries. I have only two comments (in blue).

1) I leave it to the authors to decide on the terminology to describe their structures, although I do not object calling a half of a dimer a monomer, especially since each half of the RNA dimer can have its own "function", binding to distinct fluorophores in the context of a "heterodimer". In the protein field, researchers often use the term "monomer" to describe halves of dimers even if each monomer does not have, for example, enzymatic activity on its own and requires extensive contacts and rearrangements in the dimeric state to have catalytic activity.

Reply: We appreciate the comment by the reviewer and for giving us the option to decide. We do not disagree with the referee, but we prefer the terminology used in the manuscript (protomer instead of monomer) .

13) It is difficult to see the appropriate changes in all structures because only a single validation report has been submitted with the revised manuscript. However, in the submitted report there are no improvements in the quality of the ligand structure. For example, new validation report lists 21 bond length outliers, the same number as before. It is over 40% of all bonds. While it could be more challenging to fix angle and torsion angle outliers and the library values could be disputed, the bond outliers are clearly bad outliers with values deviating significantly (~10%) from the ideal values. The authors may want to fix these outliers.

Reply: We appreciate the suggestion and agree that improvement in the ligand structure is indeed needed. To tackle this issue, we generated new restraints and coordinates files for the ligand, followed by a new refinement. Comparing the new validation report (version 3) to the previous one (version 2), we have significantly improved the quality of the ligand structure. In the new version we have the following: ligand geometry with only 2 bond length outliers in total (previously 21), 11 bond angles outliers (previously 24), and 4 torsion outliers (previously 12). The RNA structure quality remains similar to the previous report.

Reviewer #2 (Remarks to the Author):

The authors have addressed all points raised by me in a constructive manner. I am confident that the quality and clarity of the manuscript has been significantly improved through the revision. I recommend the manuscript for publication.

Reviewer #3 (Remarks to the Author):

The manuscript can now be accepted for publication.